# Community Pharmacists’ Beliefs about Suboptimal Practice during the Times of COVID-19

**DOI:** 10.3390/pharmacy10060140

**Published:** 2022-10-26

**Authors:** Lun Shen Wong, Sanya Ram, Shane Scahill

**Affiliations:** School of Pharmacy, The University of Auckland, Auckland 1010, New Zealand

**Keywords:** community pharmacy, COVID-19, human factors, patient safety, pharmacist, SHELL model, suboptimal practice

## Abstract

**Introduction:** Community pharmacies are high-performance workplaces; if the environment is not conducive to safe practice, mistakes can occur. There has been increasing demand for pharmacists during the COVID-19 pandemic as they have become integral to the response. Suboptimal practices in the work environment and with pharmacists and their teams can impact the safe delivery of services. New Zealand pharmacists’ perceptions of the current work environment and beliefs around whether suboptimal practice have increased within the last five years and the effect of the COVID-19 pandemic on their practices are unknown. **Aim/Objectives****:** To assess what New Zealand pharmacists associate with suboptimal practice in their workplace and investigate the effect of the COVID-19 pandemic on pharmacists and their workplaces. **Methods****:** We employed an anonymous online questionnaire derived from a human factors framework utilised in the aviation industry to explore the potential environment, team and organisational factors as the determinants of suboptimal work practices. The software, hardware, environment and liveware (S.H.E.L.L) model was adapted to create questions classifying the risk factors to potentially identify aspects of work systems that are vulnerable and may provide risks to optimal practice. Additional perceptions around the effect of COVID-19 on their workplace and roles as pharmacists were explored. Participants were community pharmacists working in New Zealand contacted via a mailing list of the responsible authority for the profession. **Findings****:** We received responses from 260 participants. Most participants indicated that suboptimal practice had increased in the last 5 years (79.8%). The majority of participants indicated that COVID-19 had impacted their workplaces (96%) and their roles as pharmacists (92.1%). Participants perceived that suboptimal practice was associated with a lack of leadership and appropriate management; poor access to resourcing, such as adequate staff and narrow time constraints for work tasks; a lack of procedures; competition; and stress. A lack of experience, professionalism and poor communication between staff, patients and external agencies were also issues. COVID-19 has affected pharmacists personally and their work environments. Further study in this area is required. **Conclusion****s:** We have identified that pharmacists across all sectors of New Zealand agreed that suboptimal practices had increased in the last 5 years. A human factors S.H.E.L.L framework can be used to classify themes to understand the increases in suboptimal practice and the role of COVID-19 on pharmacist practice. Many of these themes build on the growing body of the international literature around the effect of the pandemic on pharmacist practice. Areas for which there are less historical data to compare longitudinally include pharmacist wellbeing and the impact of COVID-19.

## 1. Introduction

Medicines are the most common intervention in the healthcare system. Pharmacists are the medicines experts involved in their source, supply and safe and optimal use [1,2]. Pharmacy is a high-performance workplace and if the environment is not conducive to safe practice, mistakes can occur. These mistakes can often have significant unintended consequences for consumers, wider communities and pharmacist-responsible authorities [1].

There has been an increasing demand for pharmacists during the COVID-19 pandemic as they have become integrally involved in the response. Vulnerability to workforce shortages in the health system, including pharmacists, and interruption to global supply chains for medicines and other supplies continues to challenge the system. It has also exposed the positive performance of healthcare professionals and their working environments [1,3].

Pharmacists have faced increased workloads; the COVID-19 pandemic has brought about significant challenges and pressures in the primary and secondary care environments where pharmacists have previously reported working [4,5]. In New Zealand, pharmacists were one of the most accessible healthcare professionals during the “Red Alert” COVID-19 settings. These settings limited business operations to essential services; pharmacies and access to pharmaceuticals is an essential service. During this period of restriction, pharmacies in New Zealand continued to physically see patients and provide services where access to other providers, such as General Practitioners, was limited or not face to face.

Internationally, there has been reported literature on pharmacist burnout and the effect of COVID-19 on the workforce and health system [6,7]. This paper focuses on the effects of the pandemic on community pharmacists in New Zealand.

Internationally, factors associated with the burden of COVID-19 on pharmacists include increases in the pharmacist scope, a lack of resources and staffing, excessive workloads, inefficient systems, ineffective communication, time pressures, outdated technology, financial burdens, the wearing of personal protective equipment (PPE) and its limited availability and the ever-changing infection protocol guidelines [6,8,9,10,11,12,13].

The pharmacist workforce are predominantly women, who have also reported increased stress, particularly regarding childcare arrangements, which has often meant balancing family responsibilities in addition to greater work responsibilities [14,15,16].

There is limited evidence regarding pharmacists’ mental health and reported wellbeing and the effect of sleep and protected rest periods during the pandemic [6,17,18]. There has been limited research on the role of COVID-19 and the effects of burnout and a stressed pharmacy workforce in the New Zealand context.

The international literature has described similar experiences in terms of essential service provider status [19]. Service examples included blood pressure monitoring for antihypertensives and reporting results to prescribers for review, measuring uric acid levels for gout management, providing COVID-19 vaccinations, prescribing antibiotics for urinary tract infections, providing and prescribing antiretrovirals for active COVID-19 infections, INR testing for warfarin management, participating in rapid antigen testing for local communities, providing assistance with infection control and telehealth services. Mitigating medication shortages and contributing to reporting household domestic abuse to police was also reported, which mirrored international studies [20,21,22,23].

Pharmacy work culture also affected how pharmacists worked during the pandemic. An international review [24] from the Manchester Group framed a narrative analysis around five domains of organisational culture: the professional–business role dichotomy; workload, management style, social support and autonomy; professional culture; attitudes to change and innovation; and entrepreneurial orientation. They highlight that environmental and organizational factors help shape culture and its impacts on the pharmacy workforce, services delivered and business outcomes.

Considering these issues and the proliferation of legislatively enabled, new and innovative services provided in pharmacies, it is important to identify the factors that may impact optimal practice [11,25].

Croft et al. and Weir et al. noted an increasing interest in looking at factors that affect optimal performance and which factors are involved in medication errors or incidents. A human factors framework utilised in the aviation industry and described as the “SHELL model” may be used in the healthcare sector to explore the potential environment, team and organisational factors as the determinants of optimal work practices [1]. The software, hardware, environment and liveware (SHELL) model can be used to classify risk factors to potentially identify the aspects of work systems that are vulnerable and may contribute to suboptimal practice.

Little is known about New Zealand pharmacists’ perceptions of the current work environment. This research explores New Zealand pharmacists’ beliefs around whether suboptimal practice has increased within the last 5 years and how the COVID-19 pandemic has affected their practice.

### Hypothesis/Research Questions

Aims: This study aims to investigate the following questions:

What do New Zealand pharmacists associate with suboptimal practices in their workplace?Have suboptimal practices increased in the last five years, and if so, what are the contributing factors for this?Has the COVID-19 pandemic had an impact on you as a pharmacist and your working environment?

## 2. Methods

The study was conducted using a mixed methods approach. The study collected data via an anonymous, online Qualtrics^TM^ XM survey of registered pharmacists in New Zealand. New Zealand-registered pharmacists working in the pharmacy sector who had agreed to receive questionnaires from the Schools of Pharmacy as part of their APC renewal. Pharmacists matching the sampling criteria were invited to participate by email providing a Qualtrics URL survey link. The survey was open from 1 October 2021 until 30 November 2021. Reminders were not sent, and non-responder bias was not assessed.

### 2.1. Questionnaire Design

The questionnaire was derived from a human factors framework utilised in the aviation industry to explore the potential environment, team and organisational factors as the determinants of optimal work practices [1]. Like the aviation industry, the pharmacy is a high performance–high risk workplace and so this framework is most appropriate. The software, hardware, environment and liveware (S.H.E.L.L) model was adapted to create questions classifying the risk factors to potentially identify the aspects of work systems that are vulnerable and may provide risks to optimal practice.

The questionnaire contained three sections. Section 1 and Section 2 collected demographics and perceived factors influencing optimal practice. The next sections explored aspects of the pharmacist’s working environments and their perception of what was perceived to be important in practice generally. A Likert scale to explore risk factors and to what degree they agreed or disagreed with what occurred in their current work situations. Section 3 explored the interplays between software, hardware, environment and liveware (S.H.E.L.L) [1].

The survey was piloted among eight pharmacists. Changes included gender identity questions to include non-binary individuals and those who prefer to self-describe and changing ethnicity and workplace title demographics to better reflect current practicing environment across other sectors. Additional changes were added to further clarify questions regarding the framework and applicability to different practice settings.

Quantitative data analysis was undertaken using the IBM Statistical Package for the Social Sciences (IBM SPSS version 19) analytical software. Descriptive statistics were generated to explore frequency data according to participant characteristics [26].

The qualitative analysis of open text comments made by respondents within the questionnaire was reviewed and categorised into themes using an inductive method [27]. Braun and Clarke (2006) outline the following steps, which were applied to the process:

First, outlining familiarity with data, generating codes systematically across the dataset and searching for themes to collate into subthemes. These were tabulated and reviewed in research team meetings to check consistency across the dataset. Thematic review occurred at two levels, by question/topic and by S.H.E.L.L component analysis. Colour coding was used, and circular diagrammatic representations were made based on the components of the S.H.E.L.L model. Themes were then defended through academic argument, critiqued and justified in research team meetings. The overall narrative is depicted in a grouping of master themes which outline the absolute key drivers of suboptimal practice. This overarching pictorial depiction is presented in Section 4.

### 2.2. Ethics Approval

The study was approved by the University of Auckland Human Participants Ethics Committee on 2 July 2021 for three years. Reference Number UAHPEC22669.

## 3. Results

Of the 3039 eligible pharmacists in New Zealand, 260 engaged with the survey, giving an overall response rate of 8.56%. Figure 1 shows a visual summation of factors that pharmacists associated with suboptimal practice. The overlapping intersections represent the interconnected relationships between pharmacists and the factors that they interact and engage with.

### 3.1. Participant Demographics

The majority of pharmacists (*n* = 85, 32.7%) who responded to this survey were staff pharmacists in community pharmacies, with the least from hospital pharmacists (*n* = 36, 13.8%). As shown in Table 1, more female pharmacists (*n* = 172, 66.2%) responded than male pharmacists (*n* = 68, 26.2%). The largest proportion of respondents was in the 31–35 and 51–55 years age range (*n* = 35, 13.5%). The majority of pharmacists had been in practice 30 years and more (*n* = 90, 26.9%), and 12.7% (*n* = 33) reported being in practice 6–10 years, with 11.9% having practiced 26–30 years and 10% for two years or less (*n* = 28, 10.8%).

Close to 80% (*n* = 146, 79.8%) stated that suboptimal practice had increased in the New Zealand pharmacy sector over the last five years. The majority of those who responded indicated that COVID-19 impacted them personally as a pharmacist (*n* = 164, 92.1%) and impacted their workplace (*n* = 168, 96%).

### 3.2. What Are the Factors Which Respondents Associate with SUBOPTIMAL Practice in Their Workplace?

Regarding software, most pharmacists alluded to leadership and management as a major theme as represented by quotes in Table 2. Subthemes that evolved from this theme included the ineffective management of the pharmacy and the role of effective control. Stress about financials and a lack of quality improvement and patient-centred care were also areas that contributed to suboptimal practice.

Regarding hardware, the central theme of available resources was commonly discussed as represented by the following quote:

“Not enough time to deal with the services we aim to provide. Not enough remuneration to make the work worthwhile doing. Not enough opportunities to practice services that are available. Not enough qualified staff to bear the workload” (34)

Subthemes related to this theme included issues around the lack of staffing and time constraints; a higher volume of work to staffing ratio; a lack of funding, leading to cost cutting; stock shortages; and a lack of equipment also contributed to suboptimal practice.

Regarding environment, the operating environment was a common theme as represented by the quote:

“Untidy and disorganized premises which also compromises the safety and security of staff No clear Standard Operating Procedures or procedures. Staff who deviate from standard procedures. No quality assurance or ways of changing procedures that do not work”(3)

Subthemes that relate to this theme included poor workflow; lack of standard operating procedures; mess and disorganization, in addition to poor lighting; lack of security; and no lunch or break periods. A poor or negative work culture as well as racism, errors, stress and competition were additional concepts that were reported to contribute to suboptimal practice. Taking an approach against supporting best practice was also discussed as a negative contribution towards optimal practice.

Regarding liveware, personnel was a consistent theme that came up represented by the following quote:

“Poor quality individual work, so poor attitude, knowledge, laziness, unwillingness to follow process or systems…unable to think critically and respond appropriately in challenging situations.”(58)

Subthemes that evolved from this outlined a lack of experience, passion, initiative and poor professionalism and integrity as key contributors to suboptimal practices. The approach of speed checking in place of accuracy was also cited as contributing to less optimal practice.

Regarding liveware–liveware interactions, the central theme of communication was important, as represented by sample quotes in Table 2. Subthemes that evolved from this included poor communication around incomplete prescriptions and electronic prescribing issues. A lack of team communication in handovers and phone call interruptions were also likely to contribute to a suboptimal environment. Poor communication with patients, other professional bodies, and health entities was also discussed as leading contributors to suboptimal practice.

### 3.3. Has Suboptimal Practice Increased in New Zealand over the Last 5 Years?

Table 1 outlines that 56.2% of pharmacists surveyed feel that suboptimal practice had increased across the profession in the last 5 years compared with 14.2% that disagreed. There was stronger agreement about increases in suboptimal practices in the last 5 years in those with at least 30 years of experience (23.1%) followed by those with 26–30 years of practice (11%) and those with 6–10 years of practice (10.4%). Community pharmacists (23.6%), community pharmacy owners (20.3%) and hospital pharmacists (11.5%) were in greater agreement about increases in suboptimal practice.

Regarding software, most pharmacists alluded to leadership and corporatisation as major themes. This is represented by the following quote:

“I think the way the government supports pharmacy reflects how much they value it, which leaves room for improvement. The difference in priorities for pharmacist owners and pharmacist employers makes it difficult to reach a cohesive view on how pharmacies should be run, rostering and safe staffing levels etc. With the arrival of the pandemic, the weak areas and conflicting areas existing in community pharmacies have been highlighted.”(101)

Subthemes that evolved from these major themes report a lack of leadership within the sector. In business, a change in service direction, lack of support for continuing education, stock supply issues, ownership regulations and the “roll on” effects these had for the sector. The lack of recognition for the role and importance of a pharmacist, issues relating to effective control in community pharmacy, compliance adherence relating to audits of pharmacy and practice, profitability to remain viable to provide services, competition due to deregulation within the market, marketing- and image-related views based on the competition and co-payments of funded pharmaceuticals, a lack of responsibility due to pressure on the workforce and associated competition, the role and practice of electronic prescribing, remuneration and value of staff within the workforce and the associated cost pressure of the pharmacy environment were also seen.

In relation to hardware, most pharmacists alluded to funding as being a major contributor theme to the increase in suboptimal practice as represented by the following quote:

“Increase in administration, decrease in remuneration, unwillingness from many parties to allow pharmacists to do what they do best (i.e., optimize medication related outcomes). When you do not feel supported from above and more like an underpaid ‘drone’ why should you do your best?”(80)

Subthemes that evolved from this major theme include under-resourcing in the sector from both a financial and human resource point of view, in addition to a lack of resource commitment around pharmacy equipment. A focus on regional funding leading to post-code healthcare was also discussed as a barrier linked to the funding of services.

Regarding the environment, the major theme resonating amongst pharmacists was the workload pressure they were operating in as represented by the following quote:

“Discounters have moved into the area and that means pharmacy is taking shortcuts or not able to employ the right ratio of people. No wage growth and poor practices because of competition is a driver I feel. It’s no longer enough to just be a pharmacist, somehow need to be able to sell a lot of rubbish and no(t) well research things to patients because this is what brings money in to support services and dispensary. The lack of sustainability and undercutting around us makes it hard and leads to poor decisions and cuts in good practice I think”(5)

Subthemes around workload pressures included issues of inadequate staffing in relation to the growing needs of pharmacy and the expectation and growth of new innovative services.

Some pharmacists also noted that within the leadership theme that there was a belief that:

“I do not believe there is any consideration of future proofing of the pharmacy workforce and their impact on overall health outcomes for New Zealand.”(115)

Regarding liveware, the central theme was around work culture. Subthemes around work culture included discussion around stress, anxiety, low morale and burn out of staff. Taking short cuts and rushing to complete tasks was also highlighted in addition to professional dissatisfaction and changing attitudes and expectations within workplaces as well as the addition of social media.

Regarding liveware–liveware interactions, the major theme was relationships. Subthemes around relationships include a lack of support between providers, teams and patients; the need to build more multi-disciplinary team relationships; and the need to build a different relationship with the public, providers and other health practitioners. There are also the ongoing relations regarding the generational gap within pharmacy and the existence of bullying within the profession. A summary of quotes from pharmacists is available in Table 3.

### 3.4. Has COVID-19 Had an Impact on You as a Pharmacist?

Table 1 outlines that 63.1% of pharmacists responded that COVID-19 was having an impact on them compared to 5.4% that disagreed. There was greater or more agreement reported from the female members of the workforce (66.9%) and those with 30 plus years of experience (29.9%), followed by those with 26–30 years of practice (12.4%) and those with 6–10 years of practice (11.9%). Community pharmacists (28.8%), Community Pharmacy Owners (23.2%) and hospital pharmacists (14.1%) were in greater agreement about the personal impact of COVID-19.

Regarding software, most pharmacists alluded to leadership as a major theme as represented by the following quote:

“We had to formulate our response with very little early input from either District Health Boards or our own representatives and were trying to do almost twice the number of scripts and handling telephone enquiries non-stop and trying to meet the needs of our patients, many of whom are elderly”(131)

Subthemes that evolved from the leadership theme include a lack of support from upper management within the pharmacy context and a front leadership or leadership from the “top”.

There were no identifiable issues relating to the pertaining of hardware.

Regarding the environment, pharmacists alluded to three major themes: workload, finances and the general environment they operated in. A sample quote follows:

“Overloaded in work and prescriptions and patients and doctor demands and not enough hours in the day to do it all. Generally feeling over worked and even with a rise in script numbers not being able to deliver all services in the way I would like due to time constraints and financial constrains within the pharmacy”(5)

Subthemes around these three major environmental themes included longer working hours, safety, reduced staffing and dealing with unvaccinated staff and patients. Electronic prescribing and the issues associated with this were highlighted in addition to issues around stock rationing and dealing with equipment failure. Issues around job loss, discounter pharmacies and a reduction of income leading to work sustainability were also subthemes that came through the reviewed feedback. The use of personal protective equipment (PPE) by staff, work culture due to limited interactions, communication and the implications of working from home and sick leave were also subthemes identified.

Regarding liveware, the major theme was around wellbeing and subthemes describing this include general increases in stress and anxiety, reflections on mental health and the impacts of mask wearing on practice and interactions. Additional subthemes around burnout and job changes from community pharmacy to hospital or other industries were also identified.

Regarding liveware–liveware interactions, the major theme was communication as follows:

“Prescribers not understanding changes to prescribing and having to explain to them what is required. Some prescribers are intransigent and not able to help because they are frustrated. Fewer prescriptions being dispensed because General Practitioners are not available except by phone or zoom. General Practitioner clinics closing early. General Practitioner’s difficult to get hold of to check patients prescription details. Patients calling us to get hold of General Practitioners. Patients requesting us to provide emergency supply of meds as they can’t see General Practitioners.”

Subthemes around communication include the challenges and improvement of discussion between staff, district health board funders, patients and prescribers in online consultations and between pharmacists within their own settings were signalled. A summary of quotes from pharmacists is available in Table 4.

### 3.5. Impact of COVID-19 on the Pharmacy Workplace

Table 1 outlines that nearly two thirds (64.6%) of respondents felt that COVID-19 was having an impact on their workplace compared to 2.7% who disagreed. There was stronger agreement from females (67.4%) and those with 30 plus years of experience (31.6%) followed by those with 6–10 years of practice (13.2%) and those with 26–30 years of practice (12.1%). Community pharmacists (28.7%), community pharmacy owners (24.1%) and hospital pharmacists (14.9%) were in greater agreement about the impact of COVID-19 on their workplaces.

Most pharmacists alluded to leadership as a major theme as represented by the following quote:

“The workload in terms of prescriptions doubled, we have more people becoming regulars since they can’t go away or travel. It means more resources and time but for most places without the addition of adequate staff. It’s reduced face to face communication with owners, expectations for work output haven’t changed with pandemic. Taking leave if needing to isolate or showing symptoms has been difficult, owners haven’t wanted to encourage people not to come to work”(101)

Subthemes around leadership included discussion about the general lack of planning for the COVID-19 response, contract alterations, concerns from staff not being addressed at a management level and the positive promotion of roles and service provisions within the pharmacy.

There were no identifiable issues relating to the pertaining of hardware.

Regarding the environment, pharmacist discussion centred around workload, finances and the operational environment pharmacists were in, as represented by the following:

“The majority of my staff have left. They have burned out and left pharmacy permanently. I still have a great relationship with them. I have a great new team but in each case their previous employers have been unable to find a replacement for them.”(44)

Regarding finances, a participant reported:

“Staff leave for more money to vaccinate. Reduced income from retail sales to prop up dispensary to pay fixed bills. Worrying how to keep the pharmacy open if COVID-19 enters. No extra staff for split shifts. No pharmacists around to employ because they are vaccinating and getting what pharmacists should be paid but the DHBs tell pharmacy there is no money for their contracts.”(189)

Subthemes describing these issues include remuneration, inadequate staffing, higher dispensing loads and the challenges that rise from electronic prescribing and trying to communicate with prescribers in this environment. Locum availability as well as team division while fulfilling deliveries and trying to partake in new services and reallocate resources was often discussed. The pharmacy role in COVID-19 protocol enforcement, vaccine status and misinformation correction while managing patient expectations, supply chain issues and changing the design of the pharmacy was cited frequently. Dispensing errors, the pharmacy’s viability, and job security in the face of discounters and competition were also discussed.

Regarding liveware, the major theme was around wellbeing and self-improvement, as represented by the following quote:

“Staff tired and stressed, worried they will infect those in their bubble. Childcare problems when everything is closed and they need someone to look after their kids before they can come to work.”(139)

Subthemes include job satisfaction, stress, safety and security, in addition to holiday and leave changes. Staff abuse, continuing education requirements and participation in addition to everyday Personal Protective Equipment use also featured as important subtopics that participants discussed.

Regarding liveware–liveware interactions, the major theme present revolved around work culture, as represented by the following quote:

“Management have ignored issues and efficient staff are expected to work harder to cover”(252)

Subthemes further describing this theme included generational gaps between practicing pharmacists, the mergers of pharmacy and management versus staffing expectations. A summary of quotes from pharmacists about this question is available in Table 5.

## 4. Discussion

This study aimed to understand the degree to which optimal practice was occurring in the New Zealand pharmacy sector, specifically through COVID-19 lockdowns. Registered pharmacists across all sectors were examined with a view to exploring what they associated with suboptimal practices in their workplace and if they thought it had increased over the last 5 years. The role of the COVID-19 pandemic and its impacts on pharmacists and their working environments was examined. A mixed methods survey was utilized in assessing pharmacist views on these issues.

### 4.1. Participant Demographics

The majority of pharmacist respondents were staff pharmacists in community pharmacies (*n* = 85, 32.7%), followed by community pharmacy owners (*n* = 51, 19.6%), community pharmacy managers (*n* = 26, 10%), and hospital pharmacists (*n* = 36, 13.8%). The demographics of respondents was similar to the workforce statistics reported in the Pharmacy Council of New Zealand 2021 workforce demographics [28].

### 4.2. Has Suboptimal Practice Increased in Last 5 Years?

Nearly 80% (*n* = 146, 79.8%) believed that suboptimal practice had increased in the New Zealand pharmacy sector over the last five years. This was seen in pharmacists with more years of practice, particularly over 30 years (23.1%), followed by those with 26–30 years of practice (11%). Those with 6–10 years of practice (10.4%) are labelled early career pharmacists and agreed that suboptimal practice was increasing. Major factors attributed to the increase included increasing workloads and financial pressures within pharmacies.

Increasing workloads, particularly over the pandemic period, appear in the pharmacy literature as a significant contributing stressor to suboptimal practice [6,8,9,11,12,13,16]. Johnston et al. [6] noted that, internationally, pharmacists worked on the frontlines and that, similarly to New Zealand, as other countries closed sectors of the workforce down, pharmacies and hospitals remained opened and worked throughout the pandemic period. Where staff where sick, no replacements were found. A pharmacist response indicated that due to electronic prescribing and the lack of patient presentation, there is a lack in control of every prescription presenting in the traditional physical manner; multiple electronic communications arriving at multiple times in a non-traditional sense have meant that staff are inundated with work with a lack of an ability to prioritize that work. This adds pressure on staff and, as Weir et al has observed, “corner cutting” occurs as staff prioritize speed and completion over other aspects of dispensing to try and reduce volumes of work and reduce contact time with patients. Increases in the pharmacists’ scope, while noted amongst our survey respondents as increasing work satisfaction, also increased workloads generally as pharmacists continued to see patients both at home and face to face. Insufficient staffing was a major contributor leading to suboptimal practice.

Financial pressures and the corporatization of healthcare organizations internationally is a significant change to the landscape and a scoping review has looked at processes, impacts and mediators of this [29]. Pharmacy is no different and a strong theme within this study is the effects of corporatization (discounting/retailing/supermarket pharmacies) and the resultant competition on the fiscal sustainability and workplace culture of affected pharmacies in the sector. Pharmacy owners feel under increasing financial pressure with the rise of discount pharmacy corporations. Internationally, the role of deregulation and corporatization has had an effect of pharmacy practices [30,31]. In response to this, pharmacists have indicated that owners reducing staffing levels to save on wages hugely impacts the ability of pharmacists to work at the higher levels of their scope of practice and there is a sense from the data that this goes beyond simply “suboptimal” to a scenario that is dangerous. Pharmacy ownership in New Zealand is regulated; however, this may not remain so in the future.

Figure 1 shows a visual summation of factors that evolved from the analysis, which highlights what respondents most associated with suboptimal practice. The overlapping intersections represent the interconnected relationships between pharmacists and factors that they interact and engage with.

### 4.3. Has COVID-19 Had an Impact on You as a Pharmacist?

Nearly two thirds (*n* = 164, 63.1%) of respondents reported that COVID-19 was having an impact on them personally. There was stronger agreement reported from female members of the workforce (*n* = 119 66.9%), and this could be due to “traditional” caregiving roles for younger families, which creates dual role pressure; studies also noted this factor on the effect of the pandemic on female members of the workforce [14,15,16]. Areas where there were greater impacts focused around pharmacist wellbeing and the use or availability of Personal Protective Equipment.

The impact on pharmacist wellbeing is relevant given the stress factors associated with working during a pandemic. Abu Hager et al. (2020) [32] noted in their cross-sectional study that older practitioners were more likely to suffer from psychological stress (45–54 years of age, *n* = 38, 11.8%). These are issues that collectively affect pharmacists’ ability to practice optimally. This reflects this study amongst the 46–60-year-old age group, reporting the impact of the pandemic on their pharmacist practice. Elbiddini et al. (2020) [33] noted that pharmacists were stepping into areas such as health screening and that since pharmacy was more easily accessible compared to general practice, the demand and increase in workload is likely to affect pharmacist wellbeing too.

Surveyed pharmacists reported feeling more stressed and anxious with many thinking about occupation changes and reflecting on their psychological resilience. Stressors were not confined to work but fears around safety and health and family and also the stress and availability of wearing Personal Protective Equipment. International studies confirm this and reinforce the use of Personal Protective Equipment as a stressor for pharmacists, who were not regular users prior to the pandemic [6,8,9,10,12,34].

Benson et al. (2022) [35] report that in the environment where practitioners are protecting themselves physically from infection with Personal Protective Equipment, the need to protect themselves psychologically was also a requirement in order to preserve optimal standards of healthcare provision. The COVID-19 pandemic and associated stressors are unlikely to dissipate soon, so there is a need to adapt so that body and mind are able to endure future stressors to come and build reserves of resilience.

### 4.4. Impact of COVID-19 on the Pharmacy Workplace

Nearly two thirds (*n* = 168, 64.6%) of respondents found COVID-19 had an impact on their workplace, irrespective of age, years of practice or gender. Relationships and adequate resourcing to support day-to-day work and new service proliferation were the biggest areas of discussion from the surveyed pharmacists.

In a review, Weir et al. (2020) [36] reported that, regarding communication between pharmacists and other healthcare professionals, the electronic prescribing systems caused a strain in communication and identified that divergence from compliance procedures was a common occurrence. The rollout of electronic prescribing has also changed the relationships that pharmacists have with other health practitioners. Access to the documentation and confirmation of clinical queries and the ability for pharmacists to reach out and obtain this information for informed decision making has necessitated relationship building with prescribers across the sector [37,38,39]. Communication and building more effective relationships have been highlighted by respondents as a required focus with a need for the commitment of resourcing.

The rollout of the COVID-19 vaccination programme has promoted pharmacist skills and services and, with further nurturing. could help pharmacists to establish more visible and recognized relationships with other health providers whilst building confidence to develop further multi-disciplinary care approaches given the less restrictive nature of pharmacy access during the various COVID-19 protection restrictions. While general practices are not able to check blood pressure on site, pharmacy referrals have occurred, which has opened the door for provider and patient to change their relationship with their pharmacist. This study highlights a perception that there is still a lack of understanding about the pharmacist role in healthcare, beyond the dispensing process and who bears responsibility for patient outcomes [40]. A commitment to resourcing these scenarios warrants further investigation.

### 4.5. Implications of Findings for Practice

The ability of pharmacists to continue to operate in a pandemic environment in a safe and optimal manner is finite. The risk to public safety in light of the current work conditions in a less than optimal environment should be of concern for regulators, community and professional stakeholders. Responsible authorities are responsible for public safety and want a safe environment for communities. Communities expect to receive adequate and appropriate safe care and professionals need to be able to provide this in a respectful and safe manner. Considerations to future-proof the pharmacy workforce and addressing issues around workforce, wellbeing and appropriate resourcing to ensure the continuity of care with adequate succession planning and communication could improve current pharmacy practice.

The findings have identified a gap in the literature. To our knowledge, this is the first time the S.H.E.L.L model has been used to summarize pharmacist responses to suboptimal practice in a New Zealand. The S.H.E.L.L model is an appropriate framework for analysing complex datasets. A forthcoming research paper addresses the S.H.E.L.L model in greater detail.

## 5. Limitations

There are several limitations to this study. The survey was distributed during a level 4 lockdown in New Zealand, which is positive as data was being collected when the environment was challenging. It is likely that this also impacted the response rate as anecdotal evidence and information from data collected suggests that the sector was under massive pressure. The questionnaire was emailed to 3039 eligible pharmacists and 260 engaged, giving an overall response rate of 8.56%. Although relatively low in terms of generalizability, in terms of numbers the sample aligns with expectations of other survey research undertaken in the New Zealand pharmacy sector. We acknowledge that there is a risk of sampling bias due to the way the survey was distributed: pharmacists that opted out of receiving research participations requests were not emailed a survey link [41]. Pharmacists were able to skip survey questions without a back function, so could have skipped ahead in error.

In terms of understanding the wellbeing of pharmacists, the survey did not ask about pre-existing mental health conditions. This information may have been important considering the assessment of the effect of psychological stress on the profession [33].

There was no way to differentiate responses received from community pharmacists or owners, regarding whether they were independently owned or operated franchise-based pharmacies. As such, further investigation into which sectors expressed stronger opinions about suboptimal factors for practice could enrich further reflections and recommendations on optimal pharmacy business models.

## 6. Conclusions

This study set out to understand the degree to which suboptimal practice might be occurring across all pharmacy sectors in New Zealand. There was an aim to understand the aspects of individual practice and the workplace that have been impacted and in what ways. The impact of being locked down during a COVID-19 pandemic was also explored.

Despite a low response rate, robust quantitative and qualitative analysis were able to be carried out to identify that pharmacists across all sectors of New Zealand—although particularly in the community pharmacy setting—agreed with the notion that suboptimal practices had increased in the last 5 years. The majority of those who responded indicated that COVID-19 impacted them as a pharmacist in their workplace. Drivers for this included a variety of themes that were classified using the software, hardware, environmental and liveware S.H.E.L.L model to better understand the increase in suboptimal practice.

Qualitative analysis strongly supports the notion that suboptimal practice has increased over the last 5 years and uncovered emergent themes which impact individual pharmacists and pharmacy organizations. Many of these themes build on the growing body of international literature around the effect of the pandemic on pharmacist practice. The impact of the COVID-19 pandemic was recent, hence there is limited historical data as comparison. It would be prudent for future research to focus on the impact of the pandemic on pharmacist well-being.

## Figures and Tables

**Figure 1 pharmacy-10-00140-f001:**
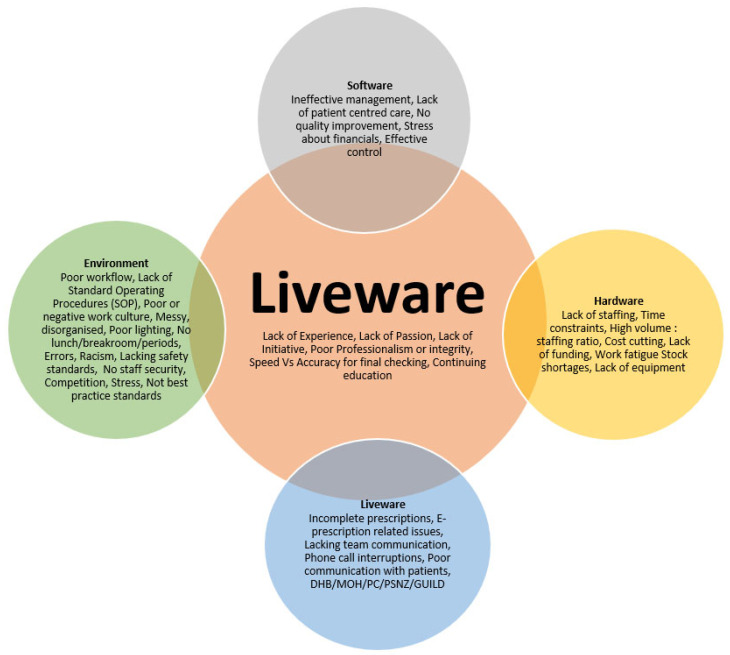
Mapped collective responses from pharmacist’s participants on factors associated suboptimal practice.

**Table 1 pharmacy-10-00140-t001:** Participant demographics.

		Has Suboptimal Practices Have Increased		Has COVID-19 Had an Impact on You as a Pharmacist		Has COVID-19 Had an Impact on Your Workplace	
Participant Characteristics	Total	Yes	No	Yes	No	Yes	No
	*n* (*n*%)	*n* (*n*%)	*n* (*n*%)	*n* (*n*%)	*n* (*n*%)	*n* (*n*%)	*n* (*n*%)
Age							
20–25 years	22 (8.5)	9 (4.9)	3 (1.6)	9 (5.1)	2 (1.1)	10 (5.7)	1 (0.6)
26–30 years	31 (11.9)	19 (10.4)	4 (2.2)	20 (11.3)	3 (1.7)	21 (12.1)	2 (1.1)
31–35 years	35 (13.5)	17 (9.3)	7 (3.8)	20 (11.3)	2 (1.1)	21 (12.1)	1 (0.6)
36–40 years	19 (7.3)	18 (9.9)	0 (0.0)	15 (8.5)	2 (1.1)	15 (8.6)	2 (1.1)
41–45 years	16 (6.2)	10 (5.5)	4 (2.2)	13 (7.3)	1 (0.6)	13 (7.5)	0 (0.0)
46–50 years	24 (9.2)	17 (9.3)	2 (1.1)	18 (10.2)	0 (0.0)	17 (9.8)	0 (0.0)
51–55 years	35 (13.5)	18 (9.9)	8 (4.4)	25 (14.1)	1 (0.6)	25 (14.4)	0 (0.0)
56–60 years	29 (11.2)	19 (10.4)	5 (2.7)	24 (13.6)	0(0.0)	24 (13.8)	0 (0.0)
61–65 years	20 (7.7)	13 (7.1)	3 (1.6)	15 (8.5)	1(0.6)	15 (8.6)	1 (0.6)
65 and above	9 (3.5)	6 (3.3)	1 (0.5)	5 (2.8)	2 (1.1)	7 (4.0)	0 (0.0)
Total	240 (92.3)	146 (56.2)	37 (14.2)	164 (63.1)	14 (5.4)	168 (64.6)	7 (2.7)
Missing	20 (0.66)	77 (29.6)		82 (31.5)		85 (32.7)	
Gender							
Female	172 (66.2)	101 (55.2)	30 (16.4)	119 (66.9)	7 (3.9)	118 (67.4)	5 (2.9)
Male	68 (26.2)	45 (24.6)	7 (3.8)	45 (25.3)	7 (3.9)	50 (28.6)	2 (1.1)
Total	240 (92.3)	146 (56.2)	37 (14.2)	164 (63.1)	14 (5.4)	168 (64.6)	7 (2.7)
Missing	20 (7.7)	77 (29.6)		82 (31.5)		85 (32.7)	
Years in Practice							
0–2 years	28 (10.8)	10 (5.5)	5 (2.7)	12 (6.8)	2 (1.1)	12 (6.9)	2 (1.1)
3–5 years	20 (7.7)	11 (6.0)	3 (1.6)	12 (6.8)	2 (1.1)	13 (7.5)	1 (0.6)
6–10 years	33 (12.7)	19 (10.4)	5 (2.7)	21 (11.9)	2 (1.1)	23 (13.2)	0 (0.0)
11–15 years	24 (9.2)	18 (9.9)	2 (1.1)	16 (9.0)	3 (1.7)	17 (9.8)	2 (1.1)
16–20 years	16 (6.2)	12 (6.6)	3 (1.6)	13 (7.3)	1 (0.6)	13 (7.5)	1 (0.6)
21–25 years	17 (6.5)	13 (7.1)	1 (0.5)	14 (7.9)	0 (0.0)	13 (7.5)	0 (0.0)
26–30 years	31 (11.9)	20 (11.0)	4 (2.2)	22 (12.4)	1 (0.6)	21 (12.1)	0 (0.0)
30 and above	70 (26.9)	42 (23.1)	14 (7.7)	53 (29.9)	3 (1.7)	55 (31.6)	1 (0.6)
Total	239 (91.9)	145 (55.8)	37 (14.2)	163 (62.7)	14 (5.4)	167 (64.2)	7 (2.7)
Missing	21 (8.1)	78 (30)		83 (31.9)		86 (33.1)	
Workplace							
Community Pharmacy Owner	51(19.6)	37 (20.3)	5 (2.7)	41 (23.2)	1 (0.6)	42 (24.1)	0 (0.0)
Community Pharmacy Manager	26 (10)	17 (9.3)	2 (1.1)	17 (9.6)	2 (1.1)	19 (10.9)	0 (0.0)
Community Pharmacist	85 (32.7)	43 (23.6)	15 (8.2)	51 (28.8)	3 (1.7)	50 (28.7)	2 (1.1)
Community Locum Pharmacist	12 (4.6)	9 (4.9)	2 (1.1)	7 (4.0)	4 (2.3)	7 (4.0)	4 (2.3)
Academia	3 (1.2)	3 (1.6)	0 (0.0)	3 (1.7)	0 (0.0)	3 (1.7)	0 (0.0)
Hospital	36 (13.8)	21 (11.5)	7 (3.8)	25 (14.1)	2 (1.1)	26 (14.9)	1 (0.6)
Industry	2 (0.8)	1 (0.5)	1 (0.5)	2 (1.1)	0 (0.0)	2 (1.1)	0 (0.0)
General Practitioner/Primary Health Organisation	10 (3.8)	6 (3.3)	2 (1.1)	8 (4.5)	0 (0.0)	7 (4.0)	0 (0.0)
Prescriber Pharm	2 (0.8)	2 (1.1)	0 (0.0)	1 (0.6)	1 (0.6)	2 (1.1)	0 (0.0)
Other	13 (5.0)	6 (3.3)	3 (1.6)	8 (4.5)	1 (0.6)	9 (5.2)	0 (0.0)
Total	240 (92.3)	145 (55.8)	37 (14.2)	163 (62.7)	14 (5.4)	167 (64.2)	7 (2.7)
Missing	20 (7.7)	78 (30)		83 (31.9)		86 (33.1)	

**Table 2 pharmacy-10-00140-t002:** Pharmacist quote sample in relation to suboptimal practice association.

What Are the Factors Which You Would Associate with SUBOPTIMAL Practice in Your Workplace?			
Framework	Emerging Theme	Sub-Theme	Pharmacist Quote
Software	Leadership and Management	Ineffective management Lack of patient centred care No quality improvement Financial Stress Effective control	“Lack of understanding by management external to pharmacy services, of what a pharmacy service is.” (71) “No quality assurance or ways of changing procedures that do not work” (3) “being badgered about money and budgets” (5)
Hardware	Resourcing	Lack of staff Time constraints Cost reduction Lack of funding Work fatigue Stock shortages Lack of equipment	“Financial pressure to churn out prescriptions while woefully understaffed due to excessive underfunding… lack of staff and therefore time with each patient” (37) “Understaffing to the point of not having enough down time to complete background tasks. Too few computers or lack of general basic equipment” (104)
Environment	Operating environment	Poor workflow and design Lack of procedures Negative work culture Poor lighting No protected breaks Errors Racism Lacking best practice Competition Stress	“Lack of breaks during day to partially recharge (e.g., a morning break or lunchtime break),” (37) “Discrimination in terms of employment opportunities by ethnicity (xenophobia) and age” (179) “poor lighting, no lunch room, untidy benches or shelves, lack of motivation from other staff” (5)
Liveware	Personnel	Lack of experience Lack of passion Lack of initiative Professionalism and integrity Speed versus accuracy Continuing education fatigue	“Lack of ethical and moral values (pharmacist and business-wise)”(3). “rushing for speed and numbers of scripts” (6) “no focus on patient counselling, not doing clinical checks on prescriptions” (19)
Liveware–Liveware	Communication	Incomplete prescriptions E-prescription related issues Lacking team communication Phone call interruptions Poor communication with patients External engagement with organisations	“Confusion between health providers regarding what has been told to the patient. Miscommunication, particularly between providers. Using jargon or terms that the patient does not fully understand. At times we have unrealistic expectations when communicating with other providers or services—there is a need for more awareness in some respects.” (235)

**Table 3 pharmacy-10-00140-t003:** Pharmacist quote sample in relation to suboptimal practice increase in the last 5 years.

Has Suboptimal Practice Had Increased in New Zealand over the Last 5 Years?			
Framework	Major Theme	Sub-Theme	Pharmacist Quote
Software	Leadership and Corporatisation	Lacking leadership Service direction Continuing education Stock Supply issues Ownership regulations Lack of recognition Effective control Compliance adherence Profitability Competition Marketing/Image Lack of responsibility Electronic prescribing Remuneration Cost pressure	“I think pharmacy owners feel they are under increasing financial pressure with the rise of discount chains and their response is often to look to the wage bill for savings. Reducing staffing levels hugely impacts the ability of pharmacists to work in the higher levels of their scope of practice and at it’s worst is more than suboptimal it is dangerous.” (32) “The pharmacy profession at large has faced a flux of change with the introduction of Corporates and the changing face of community pharmacy. This has a downstream effect on hospital practice. I have observed a loss of faith in the practical need for Post Graduate qualifications in practice and for professional development.” (230)
Hardware	Funding	Under resourcing Lack of resource commitment	“Lack of funding—this leads to us not being able to provide the services we need to as we need to spend time being money into shop. Staff unhappiness has increased due to higher work demands, less pay, more pressure: (24) “The health sector keeps wanting us to provide more services, more treatment, and store more information on site. This includes COVID-19 vaccines but there isn’t enough staff to be able to treat patients at the level that they deserve.” (40)
Environment	Workload pressure	Staffing pressure Growing needs Service provision	“Large chain pharmacies who offer “free” prescriptions but have huge wait times is putting massive pressure on their pharmacists, many of which are young and relatively inexperienced. Also COVID-19 has added an element of stress to pharmacies, increasing workload and changing how prescriptions are received. It makes it more difficult to prioritise prescriptions as you now have no idea when a person will be calling to collect their prescriptions. Customers are also more anxious and stressed and often take this out on staff members” (62)
Liveware	Work Culture	Stress and burn out Short cuts Anxiety Professional dissatisfaction Attitudes Social media	“There have been complex shifts within the age bands and ethnic diversity of the profession some have added and enriched practice whilst others have created a mismatch and muddling of shared values.” (230) “Obviously I am not a user of health services, but I have conversations with many who are. What I regard as standard level of practice is apparently not what is offered universally. I hear from patients who have seldom (if ever) had medicine or health related conversations with pharmacists. Additionally, locum’s and pharmacists I have employed have also commented on the different level of service we provide.” (119)
Liveware–Liveware	Relationships	Lack of support Multi-disciplinary teams Public Generational gap Health Providers Bullying	“Co-payments not streamlined across the board: serious ramifications including pharmacy seen as a ‘cheap supermarket service’ AND untrustworthy as causes confusion and plants suspicion in pharmacies that still need to ‘charge’ despite being a government fee. Scope of practice: as previously mentioned, pharmacists are capable of so much more as they come out of University. In part, this may be due to the cheap service that’s provided which is only heightened by the fact that pharmacies are no longer medicine-only focused. Due to the need for a retail-heavy model, trust from the general public also wanes.” (135)

**Table 4 pharmacy-10-00140-t004:** Pharmacist quote sample in relation to the impact of COVID-19 on pharmacists.

Has COVID 19 Had an Impact on You as a Pharmacist?			
Framework	Major Theme	Sub-Theme	Pharmacist Quote
Software	Leadership	Support from managementLeading from the top	“Opened opportunities to prove pharmacy is the health systems best strategic partner. Despite being late to be invited our vaccinations have dwarfed general practice efforts. Proven our worth for maintaining and increasing access to services during lockdowns where the rest of the system was in retreat. Has shifted our workforce to being mobile out of necessity via mobile vaccination services (in homes businesses and facilities and schools), as well as home deliveries for people in isolation.” (29) “Initial panic from the public.—big expectations on pharmacy teams (and a lot of gratitude from the public). Being part of the COVID vaccination programme has now prompted our District Health Board to offer Medicine Use Reviews, catchup vaccination contracts—they have seen what we can do” (134)
Hardware	N/A		N/A
Environment	Workload, Finances and General environment	Longer hours Staffing Electronic prescribing Stock ration Equipment failure Unvaccinated staff Reduced income Sustainability Job loss Discounter Safety Dealing with patients Personal Protective Equipment (PPE) Work Culture Sick leave Communication Working from home	“Increased stress levels, more time spent managing resources. (Securing products that are in short supply, double handling prescriptions that have been moved to non-stat dispensing because of supply issues, increases communication with clients around stock issues, managing customers and staff who are experiencing COVID-19 related stress, increased requirements for communication with clients about COVID-19, managing extra workload when staff are on COVID-19 related sick leave(waiting for test results after suspected COVID-19 contact or illness) Reduced human resources as a result of reduced revenue streams.” (77). “All the while government has allowed District Health Board to allow new pharmacy contracts to discounters who do no have patient care as their focus and this completely contradicts the Ministry Of Health vision/ action plan for integrative patient care... The discounters continue to drive excellent community focused pharmacies out of business at a time when patient care / need is at its highest. This drives stress at a time when profit is already low.” (44)
Liveware	Wellbeing	Stress/Anxiety Mental health Mask-wearing Job change Burnout	“Mental exhaustion dealing with so many anxious patients. Dealing with medicine shortages and supply issues. Shortage of staff due to vaccinators being paid much more and no-one to fill the gaps.” (34) “We are tired. Work is stressful. We have had to keep turning up, reading multiple repetitive emails as levels/situations change. Currently wearing masks and overseeing the public is tiring. Staff have built up annual leave. Stress is high” (112)
Liveware–Liveware	Communication	Staffing District Health Boards/Funders Patients Prescribers Online consulting Pharmacist to pharmacist	“.... also lack of stock availability and other healthcare members not pulling their weight and just referring everything to pharmacy as our doors are open to the public and we can talk to patients and see them physically is annoying” (5)

**Table 5 pharmacy-10-00140-t005:** Pharmacist quote sample in relation to the impact of COVID-19 on workplaces.

Impact of COVID-19 on the Workplace			
Framework	Major Theme	Sub-Theme	Pharmacist Quote
Software	Leadership	Concerns not addressed Lack of planning Promoting roles and services Contract alterations	“Positive—learning new skills both clinical and project management, interdisciplinary involvement, teamwork, role redesign—pharmacists and allied health vaccinators” (179) “Our workplace is shifting from relying on retail to being utilised as part of the health system. Retail has reduced by 30% however our vaccinations have sky rocketed. We are refitting our pharmacy with 5 consult rooms and removing half our retail space” (29).
Hardware	N/A		N/A
Environment	Workload, Finances and General environment	Less staff Higher dispensing load E-prescribing Deliveries Team division Misinformation correction Locum unavailability Communication with prescribers New service provision (vaccinations) Resource reallocation (consult rooms, WFH) errors Remuneration Viability Competition Discounters Job security Patient expectations Supply chain issues COVID protocol enforcement Vaccine status Pharmacy design	“Everyone is tired and stressed from dealing with misinformation and having to convince their community to get the vaccination. And then expected to perform normal Pharmacy services on top of this.” (30) “The atmosphere is less friendly as we do not want to spend a lot of time with our patients and try to get them out as safely and quickly as possible so they do not have to spend time in an at-risk area”(6) “We have had to rearrange the layout of the workplace to ensure safer environment for patients and staff.” (34) “Staff having to be off work due to testing requirements and splitting into 2 teams to give added protection for the business. This has impacted on workflow and dynamics.” (235)
Liveware	Wellbeing and self-improvement	Stress Safety/security Job satisfaction Personal Protective Equipment adornment Continuing education Holiday and leave changesAbuse	“Everyone’s ears hurt, dry hands, Hard to hear people, Physical distancing is hard for social creatures. Sanitising measures put in place constricting and constantly reminds us of COVID. Can’t go to work without thinking of COVID vaccines and how to make process less stressful” (51) “Along with fatigue, probably taking less days off as reluctant to go away” (67) “They way the pharmacy runs has drastically changed. The amount of abuse and stress that each staff members have to bear has increased.” (148)
Liveware–Liveware	Work culture	Generational Gap Mergers Management vs. Staff	“Just now they are fully staffed but all staff are really young (under 25) except for me” (45) “Initial lockdown- 2 pharmacies merged into one. Stress from other staff members and workload made it unpleasant.” (79)

## Data Availability

The data presented in this study are available on request from the corresponding author. The data are not publicly available due to ethical and privacy restrictions.

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
