# Peer review of "Community Pharmacists’ Beliefs about Suboptimal Practice during the Times of COVID-19"

_pharmacy, 2022, doi:10.3390/pharmacy10060140_

Round 1

Reviewer 1 Report

The amount of information provided in the tables is overwhelming. Is it possible to cut down on the number of quotes that represent each theme? Also the demographics table could be condensed considerably. In the demographics table, I suggest only giving the totals for each characteristic and not the Yes versus No for each of the 3 questions posed. There is just too much data here. 

Author Response

Dear Reviewer 1,

Thank you for taking time to review this paper and for your feedback.

Please see the attachment which discusses queries you have about the paper.

We hope the response answers your questions.

Thank you again for your feedback, it has strengthen the paper and our views around this paper.

Kind regards,

Lun Shen Wong on behalf of the research team Dr Sanya Ram and Associate Professor Shane Scahill

Reviewer 2 Report

The paper is interesting and describes the problems of pharmacy practice during the COVID-19 pandemia and the workload in pharmacy practice, which is quite an emerging problem nowadays. Thank you for your exciting research. Below I present my comments.

According to the aims described in verses 120-125, researchers focused on a few aspects of workload, shown in the results section; in my opinion, they do not focus mainly on COVID-19 as it is written in verses 148-149. I propose to delete the sentence "This paper focuses on responses regarding Covid-19 impacts on pharmacists and their working environments in New Zealand".

The percentage of what value is shown in brackets in table 1 is unclear. 

In the pharmacists' quote, some abbreviations exist, which in my opinion, could not be understood by readers, for example, Table 1 - MOH, CD register, Table 2 - MDT, PG, Table 3 - DHB, MUR, ED Dr, Table 5 - PPE. Authors should explain them.

Authors should change verses 417-422, which are repeated from verses 120-125.

Author Response

Dear Reviewer 2,

Thank you for taking time to review this paper and for your feedback.

Please see the attachment which discusses queries you have about the paper.

We hope the response answers your questions.

Thank you again for your feedback, it has strengthen the paper and our views around this.

Kind regards,

Lun Shen Wong on behalf of the research team Dr Sanya Ram and Associate Professor Shane Scahill

Reviewer 3 Report

Dear Authors:

Good job assessing the community pharmacists' beliefs about sub-optimal practice during the COVID-19 pandemic.

I have a few comments and suggestions for your consideration -

Generally: be consistent between the use of "COVID-19" and "Covid-19."

Abstract:

Lines 25-27: this sentence is unclear. Please redress this sentence in a way that it is understandable by a diverse audience of readers.

Please note: the "Findings" section of the abstract is too long. Restrict this section in the abstract to the most relevant findings, possibly 3 or 4 sentences, simply by answering the 3 research questions highlighted:

What do New Zealand pharmacists associate with suboptimal practices in their workplace?  Have suboptimal practices increased in the last five years and if so, what are the contributing factors for this? Has the Covid-19 pandemic had an impact on you as a pharmacist and your working environment?

Lines 39-40: This sentence mirrors exactly what was written in the conclusion section of the main article. I would recommend not repeating the same sentence, instead synthesize.

1.1. Hypothesis/Research Question -

Line 120-121: you may include a question mark (?) following the research question.

Line 124-125: COVID-19 impacted everyone in one way or the other. It may be worthwhile to qualify the impact - positive or negative.

Consider removing the personalized pronouns (you and your), so that the question can now read as: What is the impact of the COVID-19 pandemic on pharmacists and their working environment?

Methods -

Consider including a sentence or two about the study design.

Line 127: Consider including the version number of Qualtrics that was used.

Lines 155 -156: Consider including the level of significance used (for example, p<0.05)

Results -

Lines 181 -184: It may be unnecessary stating what we already have in table 1. Consider collapsing some of the numbers, and state only the obvious. For example, "The majority (n=85, 32.7%) of respondents were staff pharmacists in community pharmacies, with the least being hospital pharmacists (n=36, 13.8%).

Note: To make the results section more interesting to readers, it may suffice to state the maximum and minimum participation or figures, being significant references.

Line 194: You may rephrase as "What are the factors respondents associated with suboptimal practice in their workplace?"

Tables 2, 3,4, & 5: Some of the quotes is >2 per subtheme. Consider limiting the pharmacist quote per subthemes to 1-2. This could help improve readers' interest in your article.

Limitations -

Is there a place for response bias? I think you can include it as a limitation.

Because a section of your methods (data collection using surveys and the response rates relative to the general population) is similar to another recent study, you may draw some ideas for the limitation section there: https://www.mdpi.com/2414-6366/7/9/215/htm

Here is the reference if you need it:

Aremu TO, Singhal C, Ajibola OA, et al. Assessing Public Awareness of the Malaria Vaccine in Sub-Saharan Africa. Tropical Medicine and Infectious Disease. 2022;7(9):215. doi:10.3390/tropicalmed7090215

Conclusion -

Line 568: This is the first place "Aotearoa" is mentioned. If Aotearoa is relevant to the population considered, you may want to include it to the Abstract (line 12), Introduction, Research Question, and Method sections.

Lines 583-584: Because the pandemic was quite recent, the paucity of historical data is a known fact. You may want to conclude this section with something like this, "Because the COVID-19 pandemic was recent with less historical data, we recommend that researchers study the impact of the COVID-19 pandemic on pharmacists' wellbeing."

Author Response

Dear Reviewer 3,

Thank you for taking time to review this paper and for your feedback.

Please see the attachment which discusses queries you have about the paper.

We hope the response answers your questions.

Thank you again for your feedback, it has strengthen the paper and our views around this work.

Kind regards,

Lun Shen Wong on behalf of the research team Dr Sanya Ram and Associate Professor Shane Scahill

Round 2

Reviewer 3 Report

Dear Authors,

This looks like an improved version. Thank you for improving the quality of this manuscript by incorporating some of my suggestions.

Just a minor observation for your consideration:

Results and Discussion: Please include a comma (,) between the numbers and percentiles. For example: You have (n=85 32.7%). Include the comma to look like this: (n=85, 32.7%). There is a tone of them that needs to be updated in the results and discussion sections.